# Gender Differential Morbidity in Quality of Life and Coping Among People Diagnosed with Depression and Anxiety Disorders

**DOI:** 10.3390/healthcare13070706

**Published:** 2025-03-23

**Authors:** Elisabet Torrubia-Pérez, Maria-Antonia Martorell-Poveda, José Fernández-Sáez, Mónica Mulet Barberà, Silvia Reverté-Villarroya

**Affiliations:** 1Nursing Department, Campus Terres de l’Ebre, Universitat Rovira i Virgili, 43500 Tortosa, Spain; jfernandez@idiapjgol.info (J.F.-S.); silvia.reverte@urv.cat (S.R.-V.); 2Advanced Nursing Research Group, Universitat Rovira i Virgili, 43002 Tarragona, Spain; 3Nursing Department, Campus Catalunya, Universitat Rovira i Virgili, 43002 Tarragona, Spain; 4Unitat de Suport a la Recerca Terres de l’Ebre, Fundació Institut Universitari per a la Recerca a l’Atenció Primària de Salut Jordi Gol i Gurina (IDIAPJGol), 43500 Tortosa, Spain; 5EAP-Tortosa Oest, CAP Baix Ebre, Catalan Institute of Health, 43500 Tortosa, Spain; mmulet.ebre.ics@gencat.cat

**Keywords:** gender, anxiety, depression, psychosocial determinants, health inequalities, life quality, copying strategies

## Abstract

Background/Objectives: Psychosocial and cultural determinants have a special influence on the development, manifestation and prognosis of common mental disorders such as anxiety and depression. The objectives of this study were to define the psychosocial profile of the people most vulnerable to the development of these health problems, analyse the symptomatology and health determinants that may influence these from a gender perspective, and evaluate the quality of life and coping strategies among the adult population with this diagnosis in a rural area of Catalonia (Spain). Methods: An observational, cross-sectional, and analytical study was conducted on 180 people diagnosed with anxiety or depression. Patients completed an ad hoc sociodemographic questionnaire, the Brief Symptom Checklist (LSB-50), the Quality of Life Scale (EQ-5D-5L) and the Brief Cope Inventory (COPE-28). Results: Women aged 45–64 with a low socioeconomic profile may be more vulnerable to common mental disorders, although psychiatric symptomatology was more pronounced in men. Women were more likely to have problems with mobility (aOR= 2.93, *p* = 0.039) and daily activities (aOR = 2.75, *p* = 0.033), as well as lower self-perceived health scores (*p* = 0.002). Women used active coping, venting and seeking social support as coping strategies, while men used behavioural disengagement. Conclusions: It has been observed that the people most susceptible to developing depression and anxiety disorders may have a specific profile. Although a greater number of women have these common mental disorders, men tend to have more noticeable symptomatology. The coping strategies most used also differ according to gender.

## 1. Introduction

The latest scientific trends and the understanding of the social determinants of health locate the causes of many health problems in the interaction of the individual with biological, psychological or social spheres and focus on the impact of emotional distress in this process [1,2]. It is estimated that 280 million people worldwide suffer from depression and 301 million from anxiety, the most common mental disorders. Anxiety disorders encompass a range of conditions characterised by excessive fear or worry and related behavioural disturbances. On the other hand, people living with depression have persistent feelings of sadness, irritability, hopelessness about the future, and a lack of interest or pleasure in activities once enjoyed [2]. The number of cases of both disorders has been increasing since records were first kept, and in the development of both these, psychosocial and cultural factors, as well as other possible stressful life events to which the person may have been subjected, can act as determinants of health [2,3,4].

From an intersectional perspective, it is necessary to highlight how social, political, economic and cultural life conditions influence the development of depression and anxiety. In this regard, strong gender inequalities are a feature of the sociocultural context that conditions these health determinants [3,5]. Although the male gender is also affected by certain sociocultural demands, the hierarchy of gender-based differences makes women more vulnerable to the development of certain mental disorders [1,6]. Recent publications agree that women, compared to men, suffer more from depressive and anxious disorders, and that there are preventable determining factors [5]. The social environment exerts a strong influence on the health-disease-care process, with gender roles, stereotypes, beauty ideals and the sexual division of labour having a notable impact on different lifestyles. These can also influence how people become ill, access health services or even mortality [3,7,8].

Previous population studies in the same area in which this study was conducted have shown that the percentage of women suffering from psychosomatic disorders, including anxiety and depression, is almost double that of men (24.42% vs. 13.1%). In a rural population of 179,574, there were 8427 inhabitants diagnosed with depressive disorder (69.25% women vs. 30.75% men) and 18,311 inhabitants diagnosed with anxiety disorder (66.42% vs. 33.58%) [6]. These results are in line with similar studies assessing the same mental health problems in other areas [9,10] and also worth highlighting emotional distress (32% vs. 17.3%), which is on the rise, especially among women [11]. Globally, women are almost twice as likely as men (with a female:male ratio of 1.9:1) to develop anxiety disorders, regardless of social class [12]. Other important aspects of psychosocial and cultural factors as determinants of health relate to gender on an intersectional basis. The number of mental health diagnoses is higher among people who belong to the less-favoured social classes (14.0% of the underprivileged class vs. 4.2% of the affluent class) and among people with a lower level of education (15.2% have no or only primary education, while 5.2% have university studies) [11]. These characteristics coincide, in turn, with socioeconomic inequalities perpetuated by gender, generating hypotheses, accepted in several studies, which suggest that gender inequalities are enhanced by social inequalities [13,14]. This makes it necessary to specifically study these factors in relation to gender and the development of anxiety or depression.

Other causes of these inequalities are related to the use of time outside the workplace. Women have less free time than their male peers, and their hours away from work are more often spent on household chores or caring for dependents [15]. They also spend less time on regular physical exercise, which directly affects their quality of life [16,17]. Quality of life is a valuable indicator of the overall health of an individual, and it is usually compromised in patients with these health problems, especially among women [11,16,18]. In general terms, in rural areas like the one where this study is developed, the well-being and quality of life of the inhabitants are better than in urban areas. However, differences due to personal and health characteristics become more important, and being female and older are associated with low mental well-being [19]. The latest report from the Health Survey of Catalonia (2020) shows that nearly a quarter of the adult population suffers from emotional distress and highlights that in the rural area, where this study was carried out, there are higher rates of anxiety and depression [11].

It is important to note that the multifactorial aetiology of some common mental disorders also takes into account other individual characteristics related to coping resources [20]. The use of one or more strategies may condition the effectiveness of the body’s response to different stressful situations [21]. This also may condition resilience as a tool against adverse events, which is highly influenced by self-esteem, an aspect determined by the relationship with the environment [22]. Efficient emotional channelling or more decisive coping strategies act as protective factors, whereas an avoidant or repressive attitude contributes to the development of health problems [21,23]. According to various authors, men and women also diverge in this aspect [24,25,26]. As a result of primary socialisation, the literature shows that some men tend to express discomfort through anger, toxic habits or isolation. Instead, women seem inclined to more polarised responses, either to repress emotions to avoid problems or to seek social support in the immediate environment [15,24]. This makes a differential impact on the healthy and effective management of interaction with the context, and even the development of physical manifestations with no apparent organic origin [25,27]. Related to this, it is convenient to consider that epigenetics is also relevant to the responses. For example, hormonal factors could condition the immediate reactivity in stressful situations. In this case, it depends on the sex because of its physiological properties; however, it is just a predisposition, because life experiences can continue changing the body, perceptions, and mechanisms. It is an indeterminate disposition, not predictive [26].

People who are more exposed to adverse experiences and psychosocial and cultural inequality factors are more likely to develop anxiety or depression due to the phenomenon of intersectionality [28]. However, since by being a woman there is more exposure to these inequalities [13,28], it is essential to analyse this data considering gender as a key variable in relation to the mentioned determinants. Given the importance of these different facets in the possible development and manifestation of anxiety or depression disorders, the objectives of this study were to define the psychosocial profile of the people most vulnerable to the development of anxiety and depression disorders, analyse from a gender perspective the health determinants, analyse the psychopathological symptomatology, and evaluate the quality of life and coping strategies among adults with these common mental disorders in a rural area of Catalonia (Spain).

## 2. Materials and Methods

### 2.1. Study Design

An observational, cross-sectional and analytical study was carried out among people diagnosed with anxiety or depressive disorders in the Primary Care Area of the Catalan Institute of Health (ICS) in Catalonia, Spain.

### 2.2. Sample and Recruitment

The participants were recruited from a Terres de l’Ebre Primary Care Centre using non-probabilistic convenience sampling. Coded lists of patients eligible to be included in the sample were extracted from the e-CAP database, a computerised clinical history management tool used by Primary Care centres under the Catalan Institute of Health (ICS).

Based on the population studied by Torrubia-Pérez et al. (2022), the proportion of subjects required for this study was calculated using an estimate of a finite reference population (179,574 inhabitants) [6]. With a proportion estimate of 20% (minimum difference between men and women), a confidence interval of 95%, a precision of ±0.1 units, and the expected percentage of necessary replacements at 10%, a random sample of 175 individuals was considered sufficient. For this purpose, the GRANMO programme [29] was used.

The inclusion criteria for the subjects were to have a depression or anxiety disorder diagnosis, to be registered in the Tortosa Primary Care Center (West zone), to be over 18 years of age and not legally incapacitated, and to know how to read and write, as well as show some reading comprehension. Criteria for exclusion were not understanding Spanish, Catalan or English, presenting an advanced cognitive impairment or any other concomitant severe psychiatric problem.

Recruitment was carried out by telephone contact, through which the participants were informed about the study and provided with an online address for participation. All listed patients with depression or anxiety diagnoses were contacted. After obtaining informed consent, those who agreed to participate completed the questionnaires using Google Forms, from which the data necessary for carrying out the study were obtained. Recruitment took place between April and December 2021, and a total of 180 participants were recruited.

### 2.3. Measures

Sociodemographic variables such as gender, marital status, level of education, number of dependents, employment status, working hours and monthly salary (Minimum Interprofessional Wages as specified in the Royal Decree 231/2020 of 4 February) [30] were collected through an ad hoc questionnaire. Information was also collected on the uses of time outside the workplace, which included household chores, leisure activities, personal care, care of dependents and physical activity.

To establish the psychosocial determinants of health, manuals describing the conceptual frameworks of contextual determinants of health [31] as well as guidelines from institutions such as the World Health Organisation [32] were followed.

On the other hand, the variables and response options relating to the uses of time outside work were configured following descriptions in existing literature [1,7,15,33,34]. Based on this, the following time ranges were chosen: <15 min daily; between 15 min and 1 h daily; between 1 h and 2 h daily; and >2 h daily.

For each of the variables related to the uses of time, the tasks or activities covered by each item were specified in the same questionnaire (see Appendix A).

The ad hoc questionnaire also described physical activity levels, which were defined following the international measurement system extracted from the International Physical Activity Questionnaire (IPAQ) used by national surveys [35]. Thus, the response options were as follows: weekly ‘low’ physical activity (low activity, sedentary lifestyle), ‘moderate’ (daily walking/moderate activity 3 or more days per week, or for at least 20–30 min per day) or ‘high’ (vigorous activity at least 3 days per week, or medium intensity daily activity) levels.

#### 2.3.1. Brief Symptom Checklist (LSB-50)

The clinical instrument LSB-50 [36], or Brief Symptom Checklist, which measures psychological and psychosomatic symptoms, was used to confirm the suitability of the subjects in the sample as psychopathological patients. This is the Spanish version of the Symptom Checklist-90-R (SCL-90-R), adapted by the Institute of Psychotherapy and Psychosomatic Research [37]. It is a questionnaire made up of 50 items that identify and assess the psychological and psychosomatic symptoms of the individual who completes it. Responses to the items were collected on a five-point Likert scale (from 0 = “not at all” to 4 = “extremely”). It contains two validity scales, three global indices, nine clinical scales and subscales, and a psychopathological risk index. The results obtained from the LSB-50 questionnaire were converted according to the authors’ guidelines [36]. According to the manual, the interpretation of the questionnaire differs depending on whether or not the study population presents clinical psychopathology. Given that one of the inclusion criteria for this study was to be diagnosed with anxiety or depression disorder, the scale corresponding to a population with clinical psychopathology was applied. The scale was also adjusted according to sex (male or female), a variable that we made equivalent to gender in this study, due to the psychosocial and cultural analytical approach used.

The clarity and simplicity of the items, as well as the suitability of the questionnaire for our study population, are noteworthy. Regarding psychometric characteristics, the Cronbach’s Alpha coefficient calculated for this study was α = 0.97.

#### 2.3.2. Quality of Life Questionnaire (EQ-5D-5L)

Participants’ health-related quality of life (HRQoL) was obtained using the Euroqol-5D-5L (EQ-5D-5L) [38], which is a self-assessment index of health. The instrument comprises five dimensions—mobility, self-care, habitual activities, pain/discomfort and anxiety/depression—each with five levels of severity, and a Visual Analogue Scale (VAS) on which the subjects rate their own health status (at the moment of completion) on a scale from 0 to 100 [39].

It is an instrument used both nationally and internationally for the estimation of self-perceived health indices due to its simplicity, validity and reliability [39,40]. In terms of psychometric characteristics, the Cronbach’s Alpha coefficient calculated for this study was α = 0.71.

#### 2.3.3. Coping Strategies Questionnaire (Brief COPE-28)

Participants’ coping strategies were measured using a previously validated instrument, the Spanish version of the Brief COPE [41], adapted and translated by Morán et al. [42]. This takes the form of a 28-item multidimensional inventory that assesses different forms of response to stress. Possible answers range from 0 = “I never do this” to 3 = “I always do this”.

The calculation of the results obtained in the different responses gives rise to 14 subscales describing different possible responses to stress (active coping, planning, instrumental support, emotional support, self-distraction, venting, behavioural disengagement, positive reframing, denial, acceptance, religion, humour, self-blame and use of substances) [42]. According to their characteristics, the strategies were grouped into four blocks: cognitive coping, blocking coping (avoidance), coping through social support and spiritual coping (see Appendix A). This is a widely used, consistent and reliable questionnaire. In terms of psychometric characteristics, the Cronbach’s Alpha coefficient calculated for this study was α = 0.78.

### 2.4. Ethics Statement

This study was approved by the Research Ethics Committee of the Jordi Gol University Institute of Primary Care Research (Jordi Gol Institut Universitari d’Investigació en Atenció Primària—IDIAP Jordi Gol) with the code 20/157-P, on 1 December 2020. Prior to inclusion, participants were informed about the study and their signed informed consent was obtained.

### 2.5. Statistic Analysis

The analyses were carried out from a gender perspective, so all the data obtained were compared by gender. The variables were recoded, and descriptive statistics were used to detail the sociodemographic information of the sample using absolute frequencies and percentages. To compare sociodemographic variables by gender, the Z-test for differences of proportions was used, meaning comparison and analyses of dependence between variables were performed. The same statistical test was used for the analysis of the LSB-50 and EQ-5D-5L. For the analysis of the continuous variables, the scale of the non-parametric Mann–Whitney U test was used. So, to compare whether there were statistically significant differences in the self-perceived health VAS scale, the median and standard deviation (SD) were calculated, and this statistical test was also used. As indicated above, the psychometrics of the study sample for the LSB-50, EQ-5D-5L and COPE-28 questionnaires were calculated using Cronbach’s Alpha (acceptable values α ≥ 0.70) [43].

The results obtained from the EQ-5D-5L were dichotomised according to the analysis manual [38], and adjusted odds ratios (aOR) were applied to determine the strength of association between the presence of problems in the different dimensions of quality of life and gender. To avoid confounding variables, the OR was adjusted for age, marital status, salary, education and employment status. To analyse the results of COPE-28, which did not follow a normal distribution, medians and the 25th and 75th percentiles were used.

In all procedures, the level of statistical significance was adjusted to *p* < 0.05. Data were analysed using IBM SPSS V.28 statistical software (URV license) for Windows.

## 3. Results

### 3.1. Sociodemographic Characteristics of Sample

The final sample consisted of a total of 180 participants aged between 22 and 88 years. Of the total, 107 were diagnosed with anxiety, 51 with depression and 22 with both mental disorders. 68.33% were women, and the remaining 31.67% were men. There were no responses for “non-binary” gender. The original cases in the e-CAP programme were recorded according to the sex variable and all of them matched the gender variable, meaning that all participants were cisgender men and women.

When comparing the variables by the gender of the study subjects, statistically significant differences are found in the 25–44 and 45–64 age range groups (both *p* < 0.001). Regarding marital status, a higher percentage of women were married (51.22% vs. 45.61%, *p* < 0.001), divorced (18.70% vs. 17.54%, *p* = 0.001) and widowed (4.88% vs. 0.0%, *p* = 0.001), whereas a higher percentage of men were single (13.82% vs. 14.04%, *p* = 0.011). Differences were observed at all levels of education, with the highest percentages of women distributed among primary school educated (26.02% vs. 22.81%, *p* < 0.001), university educated (32.52% vs. 21.05%, *p* < 0.001) and those without formal education (1.63% vs. 0.0%, *p* = 0.046), while men were more concentrated in the secondary school-educated category (39.84% vs. 56.14%, *p* = 0.008) (Table 1).

There were also statistically significant differences between men and women among those who indicated that they had children under the age of 18 in their care (39.83% vs. 28.07%, *p* < 0.001) and dependents residing at home (10.57% vs. 3.51%, *p* < 0.001). The only participants who reported being primary caregivers to their grandchildren were women (1.63% vs. 0.0%, *p =* 0.014), and the majority among men had no dependents (59.65%). On the other hand, there were significantly more women in the categories of ‘employee’ and ‘unemployed’ (both *p* < 0.001) and of men in the ‘retired’ category (*p =* 0.031).

In terms of working hours, differences were found in three of the five options: full-time (*p =* 0.002), part-time (*p* < 0.001) and, again, unemployed (*p* < 0.001). Regarding salaries, a clear prevalence of women was evident in the category of those earning <950 euros per month (35.77% vs. 17.54%, *p* < 0.001). Significant differences were also obtained in the following two categories: a higher percentage of men reported earning between 951 and 1500 euros per month (35.77% vs. 49.12%, *p =* 0.008) and between 1501 and 2200 euros per month (15.45% vs. 17.54%). In the highest-paid category, although the percentages were unequal, there was no statistically significant difference (3.25% vs. 8.77%, *p =* 0.637).

In the items assessing time use, a statistically significant number of women responded, that they spend between 1 and 2 h (34.15% vs. 31.58%, *p* < 0.001) or more than 2 h (39.84% vs. 5.26%, *p* < 0.001) a day doing housework, and statistically more men spent <15 min (1.63% vs. 12.28%, *p =* 0.018) on the same. The majority of women’s responses to time spent on leisure were <15 min (26.02% vs. 19.30%, *p* < 0.001) and between 15 min and 1 h (48.7% vs. 26.32%, *p* < 0.001) per day, while men reported spending between 1 and 2 h on daily leisure activities (33.33%).

Time spent on self-care also revealed significant differences between women and men who spent >15 min per day (47.15% vs. 57.89%, *p* < 0.001), between 15 min and 1 h daily (45.53% vs. 40.35%, *p* < 0.001), and to a lesser extent, between 1 and 2 h per day (4.88% vs. 1.75%, *p* = 0.008). The majority of participants reported having no dependents (47.15% vs. 57.89%, *p* < 0.001). However, those who reported having someone to take care of were mostly women, who dedicated between 15 min and 1 h (13.01% vs. 10.53%, *p =* 0.003), 1–2 h (11.38% vs. 3.51%, *p* < 0.001), or >2 h (13.82% vs. 5.26%, *p* < 0.001) per day, as well as full-time (4.07% vs. 0.00%, *p =* 0.002) to caregiving.

Likewise, women gave up physical activity more frequently than men (13.82% vs. 5.26%, *p* < 0.001), and when they did engage in it, they reported doing so only at low (35.77% vs. 36.84%, *p* < 0.001) or moderate levels (38.21% vs. 42.11%, *p* < 0.001).

### 3.2. Adequacy of the Sample According to Psychopathological Symptomatology

After comparing proportions in the categorized tables, the results of the LSB-50 differentiated by gender can be seen in Table 2. The scores obtained ranged from 0 to 100, and the average symptomatology of psychopathological patients was between 17 and 83. Scores ≥ 84 (more symptoms than average) or ≤16 (less than average symptoms) indicated significant deviations from the typical symptomatology of the questionnaire reference population.

In terms of the general indices (GLOBAL, NUM and INT), it was observed that, of the total sample, 52.22% (56.10% of the women (n = 69) and 43.86% of the men (n = 25)) indicated a below-average symptom distress level (GLOBAL) compared to the reference psychopathological population. In number/frequency (NUM) as well as intensity (INT) of symptomatology, most participants presented average or below-average scores.

The overall number of cases that exceeded the average level of symptom distress was 10 (6.5% of women (n = 8) vs. 3.51% of men (n = 2)). The number of these symptoms (NUM) was above average in 12.2% of women and 7% of men (n = 4), and the intensity of the distress (INT) was above average in 3.25% of women (n = 4) and 3.51% of men (n = 2).

The psychopathological risk index (IRPsi, in Spanish), which in a psychopathological population indicates the adequacy of the sample, showed a greater number of average and below-average scores, with significant differences between genders in both (*p* < 0.001). In this index, 8.94% of women (n = 11) and 3.51% of men (n = 2) scored above average. Women scored similarly in the other two groupings, while men had more scores in the average group (73.68% of men, n = 42).

Regarding the clinical scales, between 4.44% and 12.78% of subjects scored above average. Separated by gender, this amounted to between 3.25% and 12.20% of women and 1.75% and 14.04% of men. No significance for above-average scores was found on any of the scales.

Of the total sample size, 58.89% scored normally on the Psychoreactivity scale (53.66% women (n = 66) vs. 70.18% men (n = 40), *p =* 0.036). Hypersensitivity (Hp) and somatisation (Sm) showed significant differences in the average (Hp *p* < 0.001, Sm *p* < 0.001) and below average (Hp *p* = 0.003, Sm *p =* 0.001) groups.

On the obsession-compulsion scale, 65.56% of the total sample fell within the average group. This was also true for Hostility (58.89%), Sleep disturbances (59.44%) and Extended sleep disturbances (50.56%). On the other hand, on the Anxiety scale, most scores were below average (51.11% of the total), and on the Depression scale, the cases were divided between below average (49.44%) and average (43.33%), with the latter showing significant differences between men and women (*p* = 0.018).

### 3.3. Quality of Life Results According to Gender

Regarding quality of life, most subjects reported no problems in the EQ-5D-5L dimensions of Mobility (n = 137), Self-care (n = 163) and Daily activities (n = 131), while in the last two dimensions (Pain/Discomfort and Anxiety/Depression) the distribution was more homogeneous (Table 3). When comparing the results of these items by gender, statistically significant differences were obtained in all of them, with a clear male prevalence. Thus, 88 women (48.89%) vs. 49 men (85.96%) reported no problems walking (*p* = 0.035); 107 women (59.44%) vs. 56 men (98.25%) reported no problems in the self-care dimension (*p* = 0.016); and 82 women (45.56%) vs. 49 men (85.96%) reported no problems carrying out their daily activities (*p* = 0.007).

Other categories in which statistically significant differences were observed between men and women were ‘severe problems in performing daily activities’ (5 women vs. 0 men, *p* = 0.036) and ‘severe pain or discomfort’ (23 women vs. 1 man, *p* = 0.002).

Regarding the participants’ self-perceived health on the VAS scale, the median responses of women were 61 (SD = 28) and men 75 (SD = 16), the median responses of women were a little lower at 65 [P25 = 50, P75 = 80] and men, coinciding, were 75 [P25 = 65, P = 90], which is a statistically significant difference (*p* = 0.002).

A logistic regression model was applied to explore the association between having a problem or not and gender, as indicated in the EQ-5D-5L analysis manual [38]. The dependent variable was created by dividing the dimensions into ‘no problems’ (level 1) and ‘any problems’ (levels 2, 3, 4 and 5), and an adjusted OR was applied (Table 4), so the differences between OR and aOR were significant. The model indicated that being a woman was a risk factor in the dimensions of Mobility (aOR = 2.93; 95% CI: 1.06–8.10; *p* = 0.039) and Daily activities (aOR = 2.75; 95% CI: 1.08–6.96; *p* = 0.033).

### 3.4. Comparing Copying Strategies Between Gender

Participants’ coping strategies, as drawn from the COPE-28, differed on the basis of gender. According to the median obtained in the different subscales, women reported using active coping (median = 4, *p =* 0.005) and acceptance (median = 4, *p =* 0.309) more frequently, while men did not report >4 in any of the coping strategies, obtaining more homogeneous results (Table 5).

Finally, differences between women and men were observed in active coping (4 vs. 3, *p =* 0.005), use of social support (3 vs. 2, *p =* 0.039), behavioural disengagement (0 vs. 1, *p =* 0.020) and venting (2 vs. 1, *p =* 0.041).

## 4. Discussion

After data analysis, the obtained results are revealed and suggest important findings to achieve the proposed objectives. With the aim of defining the psychosocial profile of the people most vulnerable to the development of anxiety and depression disorders, analysing the symptomatology and health determinants that may be mediated by gender, and evaluating the quality of life and coping strategies among adults with an anxious or depressive diagnosis in a rural area of Catalonia (Spain), the results could be divided according to the fulfilment of the previous objectives.

### 4.1. Determinants of Vulnerability to Anxiety and Depression Disorders

According to our findings through the most prevalent responses, the profile of a person vulnerable to developing these common mental disorders would be an adult woman (45–64 years), married, with secondary school education and without dependents, or underage children. At the employment level, the person would be in an active working situation as a full-time employee with an average salary of between 951 and 1500 euros per month. In terms of time use outside of work, they would devote 1 to 2 h a day to household chores, between 15 min and 1 h to leisure activities, less than 15 min a day to personal care and between 20 and 30 min to physical exercise (i.e., moderate activity). This profile is consistent with the characteristics of gender inequalities observed in other research [13,34,44] and coincides with the vulnerability factors described by the theory of intersectionality [3,28,45].

There were also differences in the distribution among adult men and women, with ages ranging from 25 to 64 years. This is in line with similar studies, and with data generated from national surveys [6,35,46]. According to the literature, these differences can also be observed in younger age groups [12,47], although we could not observe this in the present study, perhaps due to the low number of participants in that age group (n = 6). At the other end of the spectrum, some studies have reached the same conclusions in older subjects [48], although, in our results, these were not observed in the group aged over 65.

The literature indicates mixed results in terms of marital status as a determinant of mental health. In terms of gender, being married is associated with better male mental health, while it is generally a risk factor for females [49,50], although this is not the case in all studies consulted [22,49]. Although married women score higher in resilience [22], they carry more mental and emotional burden [50]. On the other hand, separated or divorced people are also more likely to develop major depression. In our study, we observed a higher prevalence of married and widowed women and single men. There was also a difference between divorced people, which coincides with studies in which divorced women obtain higher scores of depressive symptoms and show an added risk of suffering from chronic anxiety [50].

Although the trend among younger generations is towards an equal sharing of tasks [51,52], there are a significant number of cases where women and men show an unequal distribution of chores, especially in terms of informal work [34,49]. In line with this, in the sample of patients with anxious or depressive diagnoses, the number of women with children and dependents was found to be higher than that of men. 59.65% of men declared having no dependents, this being significantly their most prevalent response. Parenting and caregiving tasks have historically fallen on women, and are an important psychosocial determinant of health, with the burdens of a traditional caregiving role being a risk factor in the development of common mental disorders [1,53]. In up to 27% of cases of women with psychosomatic, anxious or depressive symptoms, the main psychosocial determinant is the burden of traditional gender roles and caregiving work [1].

Regarding social class, although the level of education most frequent in the responses was secondary-level education, women were significantly higher in the primary- and university-level groups. Gender differences are accentuated in people with basic education [33]. However, it is known that these inequalities are structurally perpetuated, so that differences can be observed in all socioeconomic groups, regardless of differences in education levels [13,54]. On the other hand, our results on job status and working hours corroborate the studies and statistical reports that confirm a feminisation of poverty. It has been reported that combining a full working day with informal and household tasks is complicated for women, who tend to then reduce formal working hours [7,55]. In Europe, in 2018, 31% of employed women were part-time workers (as opposed to 9% of men), and the unemployment rate for women in Spain was 17% compared to 13.7% for men [34]. Both conditions are reflected in our results and have previously been associated with poor mental health [13].

Along the same lines, the difficulty that some women face in continuing with higher education prevents their access to better-paid sectors and/or higher professional positions, keeping their salaries lower [56]. Although this phenomenon can be partly observed in our study sample, in the general population, more women obtain higher education (35% of women vs. 30% of men), although, paradoxically, this is not reflected in their professional careers [34]. The difficulty of maintaining a work–life balance on the one hand, and frustration in achieving life goals on the other, can, in many cases, lead to discomfort, stress and the development of various psychosomatic disorders, including anxiety and depression [20].

A study conducted in 2005 showed there were certain health determinants not captured by the National Health Survey, as a result of which differences in the situation of women and men could not be objectively assessed. These aspects included hours spent on reproductive work (household chores, caring for dependents or minors), or double the number of working hours, among others [57]. Therefore, we have given importance to these very aspects in the present study. The results obtained on the variables related to the uses of time are a snapshot of the inequalities perpetuated in this context by gender. The scarce time women reported spending on leisure, physical exercise and personal care coincides with results obtained in previous studies [7,15,34].

With regard to time spent caring for dependents, women were again shown to perform informal care tasks to a greater extent than men. It is striking that all the participants who reported being full-time carers were women.

Although women generally carry out household chores and the role of caregiver [13,33], as age increases, men do tend to take on these types of tasks to a greater extent [52]. Having more available time could influence doing housework; however, this could also be seen as a paradigm shift, initiated to some extent by the growing awareness of equality issues at the global level [54,58].

### 4.2. Psychopathological Symptomatology

Among people who have depression or anxiety, the determinants outlined above tend to be expressed in the form of general malaise or discomfort. In this study, the results obtained through the LSB-50 reveal an appropriate sample selection, given that most of the participants’ scores on the subscales fell within the average (41.11–65.56%). The number of cases that corresponded to a score above average was very low, while those who scored below average were higher than expected on some parameters. It is worth noting that many of these cases are surprising in their inconsistency with the literature. An example of this is the parameter of Hypersensitivity, which appeared in our study to be lower among women, although they are usually attributed to greater sensitivity or ease of crying [10]. The results regarding Somatization also stand out, given that in the average group, there were a high number of cases among both genders, although with a greater male presence (49.59% of women and 78.96% of men), and with women reporting less acute symptoms, which differs from other studies that have precisely evaluated such somatic manifestations without apparent organic cause [1,24,59,60].

In addition to these two clinical scales, the same trend of responses was observed in relation to depression, anxiety, global psychopathology and number of symptoms present. This suggests that, although in general there is a higher percentage of women with anxious and depressive diagnoses, men have more noticeable symptomatology. This could be related to the phenomenon of psychopathological under-diagnosis in men as a consequence of the lack of understanding of male expressions of distress as a result of socialisation or, in other words, of psychosocial and cultural determinants [5,61,62,63]. It, therefore, seems that for a man to be diagnosed with common mental disorders, the symptomatology must be more pronounced; otherwise, it is likely to go undetected. On the other hand, there may be a greater predisposition or more conscientious recording of the detection and diagnosis of this type of pathology in women.

### 4.3. Dimensions of Quality of Life and Self-Perception of Health Status

As we have seen throughout this discussion, psychosocial and cultural determinants shape the mental and physical health of men and women through multiple gender norms. Even at a young age, psychological distress becomes more pronounced in women precisely because of the negative associations arising from social constructs [47]. Quality of life is a fundamental indicator of the global health status of men and women, particularly, and for the purpose of this study, of their mental health [16]. In the responses obtained in our study, there was a clear prevalence in the absence of problems in the dimensions of mobility and daily activities among men. This partly coincides with previous studies at the national level in which the percentages of the values obtained for men are similar [35,40], although this is not the case for women, who showed much lower levels in our study. In both dimensions mentioned, only between 45.56 and 59.44% of women reported having no related problems. This means that at least 40% of the women in our sample do have problems with mobility or daily activities that may affect their quality of life. Furthermore, when analysing the dimensions of self-care, pain/discomfort and anxiety/depression, we see that the figures relating to the presence of problems increase exponentially in both genders, but more markedly among women. This is supported by the literature and the latest statistical reports both at the national level and in the area studied [11,35,64]. According to the latest reports in Catalonia, about a quarter of the adult population suffers from emotional distress (27.9–32% women and 17–17.3% men), and this trend is increasing in both genders, although in a higher percentage among women [11,65]. In addition, the responses that rated pain/discomfort as severe in this dimension were highly significant in women, as was the case in another recent study covering 59 countries that reported a clear gender gap in health [8].

However, our study was not able to establish a direct association between gender and the presence of problems in the dimensions of self-care, pain/discomfort and anxiety/depression. This is probably due to the characteristics of the women and men in our sample, who were, in fact, required to have these diagnoses in order to be included in the study. Surveys of the general population in the same territory and in other geographical areas have found a relationship between the female gender and these dimensions [8,11,64,65,66]. In our results, an important association was obtained on these two dimensions: women were more than twice as likely to suffer from mobility problems and more than twice as likely to suffer from a problem in carrying out daily activities. These predictive values are supported by studies that used the same instrument, given that problems in terms of mobility [11,64,66] and daily activities [64,65], have previously been registered as higher among women.

Finally, participants’ self-rated health status as assessed by the VAS scale provides significant information on the gender difference. The data indicate that women have more heterogeneous perceptions, but with a substantially lower median than men (65 vs. 75). This is interpreted to mean that, in this study, men’s self-rated health status is more positive and that women have a worse self-rated state of health, in line with other studies [8,67]. In accordance with our findings, the latest Catalan Health Interview Survey (2021) carried out in the general population, shows the same score for men (75.3) and a slightly higher score for women (70.8). The survey also reveals lower scores for people with primary education levels, of lower social class and older ages, and the female gender scores are always below those of the male gender [64].

### 4.4. Gender-Based Coping Strategies

Psychosocial and cultural determinants, such as those described in this paper, can act as both risk and protective factors. Social support networks or a sense of belonging also act as modulators, not only prior to the development of pathologies but also in influencing their prognosis [68].

There are important gender differences in expressing oneself and socialising. This study showed a tendency of women to adopt active coping strategies, vent and seek social support, while men showed significantly higher rates of behavioural disengagement. This is consistent with a study conducted with men and women of five different ethnicities, which showed that men tended to express positive and negative emotions less frequently and less intensely than women. On the other hand, women tended to be more direct in their emotions and to resort more frequently to social support [69].

The results obtained among men suggest a propensity to behavioural disengagement, which concur with previous studies, which conclude that avoidance is common in the male population [24]. Surprisingly, however, there were no higher scores on substance use, given that these strategies are often the result of social customs [15,70]. On the other hand, we did find that active coping is the most used resource among women, which coincides with existing literature [24,27]. The perception is that women are more likely to be expressive, communicate their problems and try to seek support from their close networks [27,71]. This is a protective factor for women since emotional expression can be a channel for distress and lead to a substantial improvement in health [4].

Knowledge of the different ways in which depression and anxiety may be expressed would be a step forward for health professionals in detecting the same in both genders. In addition, such symptoms are highly prevalent and often occur concomitantly with other health problems [2,6,16]. Thus, their detection is greatly positive and does not overburden health professionals or increase resource use [72], making it worthwhile to continue working on this issue in order to offer a comprehensive and individualised approach to health care.

### 4.5. Study Limitations

This study contains some limitations, such as the difficulty in the detection and diagnosis of cases among men (as discussed above), which could bias results analysed by gender. Also, having used the LSB-50, it was possible to establish the range of psychopathology of the sample and to reinforce the fulfilment of the clinical inclusion criteria. Convenience sampling was applied, which may have conditioned the profile of the participants. On the other hand, the authors consider that this study is important in terms of its location given the characteristics of rural areas in Catalonia, which currently present worrying data in terms of demography, economic conditions and accessibility to health services. It was not registered whether the patients received any kind of treatment or health interventions. Finally, the use of self-administered questionnaires as data collection instruments limits the justification of the findings and the knowledge of the particular cases of participants. Although the participants had a gap in explaining their health-related circumstances, the volume of responses was quite low. This means that information about contextual experiences was not collected, such as the COVID-19 pandemic, for example, which probably had an impact on the emotional distress of the participants. Also, a transversal observational study implies that causality of the findings cannot be established. Therefore, we propose further qualitative studies that would allow for a more in-depth understanding of the context of people with anxious and depressive symptoms.

## 5. Conclusions

Our results suggest that gender is remarkably important in the development and expression of different aspects related to anxiety and depression disorders. It has been observed that the people most susceptible to developing these common mental disorders may have a specific profile. They are adult women (between 45 and 64 years old), married, with intermediate education and without dependents or minor children in their care. At the employment level, they tend to work full-time as employees and earn an average salary of between 951 and 1500 euros per month. In terms of their distribution of time outside of work, they spend between 1 and 2 h a day on household chores, between 15 and 60 min on leisure activities, less than 15 min on personal care, and between 20 and 30 min on moderate physical exercise. The findings are consistent with the literature, and this profile coincides with the gender inequalities observed in other studies.

The study has also shown that, although a greater number of women have these common mental diagnoses, men tend to have more noticeable symptomatology. However, while the psychopathological symptoms in women appear fewer, their quality of life is also lower and appears to be affected to a greater extent. This translates into women being almost three times as likely to suffer from mobility problems and more than twice as likely to suffer from problems in carrying out daily activities and up to nine times more likely to have self-care issues.

Finally, the coping strategies most used by people with an anxious or depressive diagnosis also differ according to gender. Women tend to use active coping, venting and seeking social support as coping strategies, while men are significantly more likely to show behavioural disengagement. This indicates that coping is more problem-focused in the female gender.

The findings of this study highlight the importance of individualised care and the need to focus on the psychosocial context of the people in need of care. Thus, the ideal way to explore the different determinants of health would be an interdisciplinary approach with a gender perspective.

## Figures and Tables

**Table 1 healthcare-13-00706-t001:** Comparison of sociodemographic characteristics of the participants.

		Total	Women		Men		
		n	n	%	n	%	*p*
Total		180	123	68.33	57	31.67	<0.001 *
Age	18–24 years	6	4	3.25	2	3.51	0.248
	25–44 years	63	44	35.78	19	33.33	<0.001 *
	45–64 years	91	62	50.41	29	50.88	<0.001 *
	>65 years	20	13	10.57	7	12.28	0.058
Current marital status	Married	89	63	51.22	26	45.61	<0.001 *
	In a relationship	25	13	10.57	12	21.05	0.777
	Separated	2	1	0.81	1	1.75	1.000
	Divorced	33	23	18.70	10	17.54	0.001 *
	Sigle	25	17	13.82	8	14.04	0.011 *
	Widowed	6	6	4.88	0	0.00	0.001 *
Educational status	Primary	45	32	26.02	13	22.81	<0.001 *
	Secondary	81	49	39.84	32	56.14	0.008 *
	Superior	52	40	32.52	12	21.05	<0.001 *
	None	2	2	1.63	0	0.00	0.046 *
Family caregiver	Children < 18 years	65	50	39.83	16	28.07	<0.001 *
	Children > 18 years	5	3	2.43	2	3.51	0.527
	Children < 18 years + Children > 18 years	1	1	0.81	0	0.00	0.157
	Children < 18 years + Dependents residing at home	5	3	2.44	2	3.51	1.000
	Dependents residing at home	15	13	10.57	2	3.51	<0.001 *
	Dependents residing outside the home	1	0	0.00	1	1.75	0.157
	Primary caregiver of grandchildren	3	3	1.69	0	0.00	0.014 *
	No dependents	85	51	41.46	34	59.65	0.009 *
Occupation	Employee	103	70	56.91	33	57.89	<0.001 *
	Self-employed	12	8	6.50	4	7.02	0.102
	Unemployed	26	21	17.07	5	8.77	<0.001 *
	Student	4	2	1.63	2	3.51	1.000
	Retired	35	22	17.89	13	22.81	0.031 *
Employment status	Full-time	86	53	43.09	33	57.89	0.002 *
	Part-time	24	22	17.89	2	3.51	<0.001 *
	Moonlighting	5	3	2.44	2	3.51	0.527
	Eventual or weekends	2	1	0.81	1	1.75	1.000
	Unemployed	63	44	35.77	19	33.33	<0.001 *
Monthly salary	<950 €	54	44	35.77	10	17.54	<0.001 *
	951–1500 €	72	44	35.77	28	49.12	0.008 *
	1501–2200 €	29	19	15.45	10	17.54	0.018 *
	>2200 €	9	4	3.25	5	8.77	0.637
Daily time spent on household chores	<15 min	9	2	1.63	7	12.28	0.018 *
	15 min–1 h	59	30	24.39	29	50.88	0.854
	1–2 h	60	42	34.15	18	31.58	<0.001 *
	>2 h	52	49	39.84	3	5.26	<0.001 *
Daily time spent on leisure activities	<15 min	43	32	26.02	11	19.30	<0.001 *
	15 min–1 h	75	60	48.78	15	26.32	<0.001 *
	1–2 h	38	19	15.45	19	33.33	1.000
	>2 h	24	12	9.76	12	21.05	1.000
Daily time spent on personal self-care	<15 min	91	58	47.15	33	57.89	<0.001 *
	15 min–1 h	79	56	45.53	23	40.35	<0.001 *
	1–2 h	7	6	4.88	1	1.75	0.008 *
	>2 h	3	3	2.44	0	0.00	0.014
Daily time spent caring of dependents	No dependents	91	58	47.15	33	57.89	<0.001 *
	<15 min	26	13	10.57	13	22.81	1.000
	15 min–1 h	22	16	13.01	6	10.53	0.003 *
	1–2 h	16	14	11.38	2	3.51	<0.001 *
	>2 h	20	17	13.82	3	5.26	<0.001 *
	Caring for a dependent person full time	5	5	4.07	0	0.00	0.002 *
Weekly physical activity	None	20	17	13.82	3	5.26	<0.001 *
	Low	65	44	35.77	21	36.84	<0.001 *
	Moderate	71	47	38.21	24	42.11	<0.001 *
	High	24	15	12.20	9	15.79	0.083

Z-test; Significant at *p* < 0.05 (*).

**Table 2 healthcare-13-00706-t002:** Differences between women and men in the scales and indices of the LSB-50 in psychopathological patients.

		Total		Women		Men		
		n	%	n	%	n	%	*p*
Total	Punctuation	180		123		57		
Psychoreactivity	≤16	63	35.00	47	38.21	16	28.07	0.185
	17–83	106	58.89	66	53.66	40	70.18	0.036 *
	≥84	11	6.11	10	8.13	1	1.75	0.097
Hypersensitivity	≤16	92	51.11	72	58.54	20	35.09	0.003 *
	17–83	76	42.22	41	33.33	35	61.40	<0.001 *
	≥84	12	6.67	10	8.13	2	3.51	0.248
Obsession-compulsion	≤16	49	27.22	37	30.08	12	21.05	0.206
	17–83	118	65.56	75	60.98	43	75.44	0.057
	≥84	13	7.22	11	8.94	2	3.51	0.190
Anxiety	≤16	95	52.78	71	57.72	24	42.11	0.051
	17–83	74	41.11	43	34.96	31	54.39	0.014 *
	≥84	11	6.11	9	7.32	2	3.51	0.321
Hostility	≤16	66	36.67	47	38.21	19	33.33	0.528
	17–83	106	58.89	72	58.54	34	59.65	0.888
	≥84	8	4.44	4	3.25	4	7.02	0.254
Somatization	≤16	60	33.33	51	41.46	9	15.79	0.001 *
	17–83	106	58.89	61	49.59	45	78.95	<0.001 *
	≥84	14	7.78	11	8.94	3	5.26	0.391
Depression	≤16	89	49.44	66	53.66	23	40.35	0.097
	17–83	78	43.33	46	37.40	32	56.14	0.018 *
	≥84	13	7.22	11	8.94	2	3.51	0.190
Sleep disturbances	≤16	50	27.78	37	30.08	13	22.81	0.311
	17–83	107	59.44	71	57.72	36	63.16	0.490
	≥84	23	12.78	15	12.20	8	14.04	0.731
Extended sleep disturbances	≤16	78	43.33	58	47.15	20	35.09	0.129
	17–83	91	50.56	58	47.15	33	57.89	0.180
	≥84	11	6.11	7	5.69	4	7.02	0.730
IRPsi	≤16	84	46.67	71	57.72	13	22.81	<0.001 *
	17–83	83	46.11	41	33.33	42	73.68	<0.001 *
	≥84	13	7.22	11	8.94	2	3.51	0.190
GLOBAL	≤16	94	52.22	69	56.10	25	43.86	0.126
	17–83	76	42.22	46	37.40	30	52.63	0.054
	≥84	10	5.56	8	6.50	2	3.51	0.414
NUM	≤16	78	43.33	59	47.97	19	33.33	0.065
	17–83	83	46.11	49	39.84	34	59.65	0.013 *
	≥84	19	10.56	15	12.20	4	7.02	0.293
INT	≤16	95	52.78	67	54.47	28	49.12	0.504
	17–83	79	43.89	52	42.28	27	47.37	0.522
	≥84	6	3.33	4	3.25	2	3.51	0.929

IRPsi = Psychopathological Risk Index (IRPsi, by its Spanish acronym), GLOBAL = Global Severity Index, NUM = Number of suffered symptoms, INT = Intensity Index of suffered symptoms. LSB-50 punctuation: ≤16 = less than average symptoms, 17–83 = average typical symptomatology of the questionnaire reference population, ≥84 = more symptoms than average. Z-test; Significant at *p* < 0.05 (*).

**Table 3 healthcare-13-00706-t003:** Differences in the dimensions of EQ-5D-5L between patients.

		Total	Women		Men		
		n	n	%	n	%	*p*
Total	Items	180	123		57		<0.001 *
Mobility	No problems	137	88	48.89	49	85.96	0.035 *
	Slight problems	26	21	11.67	5	8.77	0.141
	Moderate problems	13	10	5.56	3	5.26	0.489
	Severe problems	4	4	2.22	0	0.00	0.169
	Unable to walk about	0	0	0	0	0	
Self-care	No problems	163	107	59.44	56	98.25	0.016 *
	Slight problems	10	9	5.00	1	1.75	0.130
	Moderate problems	6	6	3.33	0	0.00	0.090
	Severe problems	1	1	0.56	0	0.00	0.495
	Unable to wash or dress	0	0	0	0	0	
Daily activities	No problems	131	82	45.56	49	85.96	0.007 *
	Slight problems	24	19	10.56	5	8.77	0.220
	Moderate problems	14	12	6.67	2	3.51	0.145
	Severe problems	9	9	5.00	0	0.00	0.036 *
	Unable to do usual activities	2	1	0.56	1	1.75	0.575
Pain or Discomfort	No pain/discomfort	55	34	18.89	21	36.84	0.213
	Slight pain/discomfort	61	36	20.00	25	43.86	0.054
	Moderately pain/discomfort	39	29	16.11	10	17.54	0.361
	Severely pain/discomfort	24	23	12.78	1	1.75	0.002 *
	Extreme pain/discomfort	1	1	0.56	0	0.00	0.495
Anxiety or depression	Not anxious/depressed	50	32	17.78	18	31.58	0.438
	Slight anxious/depressed	68	44	24.44	24	42.11	0.415
	Moderately anxious/depressed	50	39	21.67	11	19.30	0.084
	Severely anxious/depressed	10	7	3.89	3	5.26	0.907
	Extremely anxious/depressed	2	1	0.56	1	1.75	0.575
VAS Scale [0–100] ^a^			Women ^b^		Men ^b^		
		180	65 [50, 80]		75 [65, 90]		0.002 * (U = 2525.5)

Z-test. ^a^ Mann–Whitney U Test. ^b^ Median [P25, P75] * Significant at *p* < 0.05.

**Table 4 healthcare-13-00706-t004:** Logistic regression model with grouped dimensions.

		OR	CI 95%		*p*	aOR	CI 95%		*p*
Any Mobility problems	Men	1				1			
	Women	2.43	1.04	5.66	0.039 *	2.93	1.06	8.10	0.039 *
Any self-care problems	Men	1				1			
	Women	8.37	1.08	64.78	0.042 *	7. 07	0.83	60.14	0.074
Any usual activities problems	Men	1				1			
	Women	3.06	1.32	7.06	0.009 *	2.75	1.08	6.96	0.033 *
Any pain or discomfort	Men	1				1			
	Women	1.52	0.78	2.97	0.214	2.24	0.99	5.03	0.051
Any anxiety or depression	Men	1				1			
	Women	1.31	0.66	2.61	0.439	1.13	0.52	2.44	0.757

aOR: Adjusted Odds Ratio; CI 95%: Confidence Interval; Significant at *p* < 0.05 (*).

**Table 5 healthcare-13-00706-t005:** Copying strategies differences according to gender.

	Women	Men		
	Median ^b^		U	*p*
Active Coping	4 (3, 5)	3 (2, 4)	2623.5	0.005 *
Planning	3 (2, 4)	3 (2, 4)	3409.5	0.763
Emotional Support	2 (2, 4)	2 (2, 3)	3129.0	0.234
Instrumental Support	3 (2, 4)	2 (1, 4)	2854.5	0.039 *
Religion	1 (0, 2)	0 (0, 2)	2949.5	0.065
Positive Reframing	3 (2, 4)	2 (2, 4)	3023.0	0.131
Acceptance	4 (2, 5)	3 (2, 4)	3181.5	0.309
Denial	1 (0, 2)	1 (0, 2)	3175.0	0.284
Humour	2 (0, 3)	2 (0, 3)	3346.0	0.616
Self-distraction	3 (2, 4)	2 (2, 4)	3022.5	0.131
Self-blame	2 (1, 3)	2 (1, 3)	3227.5	0.382
Behavioural disengagement	0 (0, 2)	1 (0, 2)	2800.5	0.020 *
Venting	2 (1, 3)	1 (1, 3)	2858.5	0.041 *
Use of substances ^c^	0 (0, 0)	0 (0, 0)	3220.5	0.199

Mann–Whitney U Test.; ^b^ Median (P25, P75); ^c^ Alcohol or drugs; Significant at *p* < 0.05 (*).

## Data Availability

The datasets used and/or analysed during the current study are available from the authors on reasonable request.

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
