# Peer review of "Gender Differential Morbidity in Quality of Life and Coping Among People Diagnosed with Depression and Anxiety Disorders"

_healthcare, 2025, doi:10.3390/healthcare13070706_

Round 1
Reviewer 1 Report
Comments and Suggestions for Authors
Dear Authors,
I congratulate you on your work on gender differences in quality of life and coping among people diagnosed with anxiety or depression disorders, and living in rural areas of Catalonia (Spain).
I have some questions and clarifications:
Title: ”MORBIDITY” or ”MOBILITY”??? Results show mobility and daily activity differences between men and women.
Introduction: what are the scientific and diagnosis differences between anxiety and depression???
Methodology: statistical analysis tests describe deeper! The subchapter 2.5. was too short!!! Must insert descriptions and conditions of statistical analysis in use.
A chi-square test for “comparison of proportion” of what??? (line 223).
Chi-square test statistical conditions???
Non-parametric test statistical conditions???
Logistic regression statistical conditions???
Results:
3.1. is a sociodemographic characteristics description. Delete p from parentheses!!!
Are there errors in your chi-square test application and interpretation!!! Different subgroups must be created as in the logistic regression model. Every subgroup created needs 5 independent respondents.
Table 1 can be a contingency table but NOT for comparisons. Delete p from the tables.
Tables 2 and 3 ARE for comparison of psychopathological ranges. Delete % from tables!!!
Insert the U value for variables in Tables 2 and 3.
Logistic regression must mention Exp (B).
I wish you all the best!
Author Response
Dear Reviewer 1,
We sincerely appreciate the time and effort you have dedicated to reviewing our manuscript. Your insightful comments and suggestions have been extremely valuable in improving the quality and clarity of our work.
Below, we provide detailed responses to each of your observations, addressing your concerns and implementing the necessary revisions accordingly.
We are grateful for your constructive feedback and hope that the modifications enhance the manuscript to your satisfaction.
Comment 1: Title: ”MORBIDITY” or ”MOBILITY”??? Results show mobility and daily activity differences between men and women.
Response 1: Dear reviewer, the title discusses the pathological differences in the problems studied as a whole. Mobility is only one parameter within the quality of life index, and we didn't want to focus on that in the title. We hope we've given you a good answer. Thank you very much for your feedback.
Comment 2: Introduction: what are the scientific and diagnosis differences between anxiety and depression???
Response 2: We agree with the you that this information was missing, so we have added a brief description in the introduction section (line 45-49). Furthermore, we have updated some information and added more context to the introduction.
Comment 3: Methodology: statistical analysis tests describe deeper! The subchapter 2.5. was too short!!! Must insert descriptions and conditions of statistical analysis in use.
A chi-square test for “comparison of proportion” of what??? (line 223).
Chi-square test statistical conditions???
Non-parametric test statistical conditions???
Logistic regression statistical conditions???
Response 3: Thank you very much for your observations. Taking into account your comment, the authors have, accordingly, modified the subchapter 2.5. So we have added the statistical test used in each instrument (lines 243-252). We have taken the opportunity to specify that the comparison of proportions was carried out using the Z-test.
On the other hand, thank you for your questions regarding the statistical conditions of the tests used in our study. We confirm that all statistical assumptions for the Z- test (use of categorical variables and ensuring that the expected frequency in each cell was sufficient), Mann-Whitney U test (two independent groups when the normality assumption was not met. The non-parametric nature of the test ensures its applicability regardless of the distribution of the data), and logistic regression (model assumptions were verified, including the absence of multicollinearity among the independent variables and the linearity of continuous variables with respect to the logit transformation, and model fit and goodness-of-fit measures were also evaluated) were met, following standard methodological guidelines. These conditions were assumed as part of the analysis, ensuring the validity of the results. However, they were not explicitly stated in the manuscript, as they are fundamental considerations inherent to these statistical methods.
We hope this explanation addresses your query. Please let us know if further details are required.
Comment 4: Results: 3.1. is a sociodemographic characteristics description. Delete p from parentheses!!!
Are there errors in your chi-square test application and interpretation!!! Different subgroups must be created as in the logistic regression model. Every subgroup created needs 5 independent respondents.
Table 1 can be a contingency table but NOT for comparisons. Delete p from the tables.
Tables 2 and 3 ARE for comparison of psychopathological ranges. Delete % from tables!!!
Insert the U value for variables in Tables 2 and 3.
Logistic regression must mention Exp (B).
I wish you all the best!
Response 4: The authors agree in your comment about the Table 1, and we have decided to modify the table title and add the comparison detail to keep de p values. The analysis carried out through comparison has contributed to the achievement of the objective of tracing the vulnerability profile of the studied population. Thank you for pointing this out to improve the presentation. Besides, your comment has made us think that perhaps it would be more appropriate to specify that the z-test was used and not define it as the chi-square test, since each category was analyzed and the former would be more correct. The reason is that in 2x2 tables the chi-square test is equivalent to the z-test, but when analyzing more variables, the z-test is more precise, which is why it was used.
On the other hand, when we designed the study and prepared the tables and results, we wanted to maintain the format of variables in rows and gender in columns because this is the key objective of the analysis in this study. The authors did not consider it appropriate to create subgroups because it is not part of the objective of the study, and the presentation of the table would be unclear and overly informative. However, we welcome your suggestions so we can explore the option of conducting a more comprehensive analysis of the data in the future.
Regarding Table 2, percentages are important information for us in the analysis, as the scores are stratified by symptom intensity. This helps in the analysis and clarity of the results by gender, that is the reason why we insert the percentages.
About the U values, attending to your request, we have added the U values in Tables 3 and 5, in which we have used the Mann-Whitney U test.
The authors greatly appreciate your feedback and believe your comments have greatly improved the original manuscript. Thank you very much for your work.
Sincerely,
The authors.
Reviewer 2 Report
Comments and Suggestions for Authors
Thanks to the authors for sharing their manuscript. In general, I like both the study and the manuscript, but I have a number of comments.
Abstract
The abstract lacks a conclusion, it "breaks off" on the results of the study.
Introduction
The introduction does not explicitly describe the scientific novelty of the research. The topic has been studied quite well, so the authors should describe in more detail what is new in this area they will explore.
Materials and Methods
As far as I know, the research conducted using the EQ-5D-5L needs to be re-registered. If the authors did this, then it is worth specifying the number of the re-registration protocol.
Results
The text in section 3.1 « Sociodemographic characteristics of sample» significantly duplicates the data presented in Table 1. It is better to remove numeric values from the text.
I consider all comments except the one about the introduction to be insignificant. When finalizing the introduction, I would recommend the manuscript for publication.
Author Response
Dear Reviewer 2,
We sincerely appreciate the time and effort you have dedicated to reviewing our manuscript. Your insightful comments and suggestions have been extremely valuable in improving the quality and clarity of our work.
Below, we provide detailed responses to each of your observations, addressing your concerns and implementing the necessary revisions accordingly.
We are grateful for your constructive feedback and hope that the modifications enhance the manuscript to your satisfaction.
Comment 1: Abstract. The abstract lacks a conclusion, it "breaks off" on the results of the study.
Response 1: Dear reviewer, you are right. We have added the subheadings within the abstract and developed the conclusions section (row 31). Thank you for pointing this out.
Comment 2: Introduction. The introduction does not explicitly describe the scientific novelty of the research. The topic has been studied quite well, so the authors should describe in more detail what is new in this area they will explore.
Response 2: Thank you for your valuable feedback. We have tried to highlight the importance of the topic in the introduction. However, we understand that this may not have been sufficiently clear. In response to your comment, we have added additional text to further emphasize the significance of our study (rows 53, 81, 125). We hope that these revisions help to better convey the relevance and impact of our research.
Comment 3: Materials and Methods. As far as I know, the research conducted using the EQ-5D-5L needs to be re-registered. If the authors did this, then it is worth specifying the number of the re-registration protocol.
Response 3: Thank you for your thoughtful comments. Regarding the registration code for the questionnaire used in our study, we would like to clarify that this registration was not carried out because, at the time of the study, it was not mandatory. Additionally, the instrument was freely accessible for download and use. Since our study required the scale to remain unaltered in any way, we did not consider registration necessary. Furthermore, the authors believe that requesting a retrospective registration of the scale would not add value to the study. We appreciate your feedback and remain available for any further clarifications.
Comment 4: Results. The text in section 3.1 « Sociodemographic characteristics of sample» significantly duplicates the data presented in Table 1. It is better to remove numeric values from the text.
Response 4: After considering your comment, we appreciate your observation nevertheless the authors have decided that it is important to highlight in the presentation of results the values ​​with statistically significant differences obtained in the analysis. The reason is because the analysis carried out through comparison has contributed to the achievement of the objective of tracing the vulnerability profile of the studied population.
The authors greatly appreciate your feedback and believe your comments have greatly improved the original manuscript. Thank you very much for your work.
Sincerely,
The authors.
Reviewer 3 Report
Comments and Suggestions for Authors
Review report: “Gender differential morbidity in quality of life and coping among people diagnosed with depression and anxiety disorders”
Highly esteemed authors Elisabet Torrubia-Pérez , Maria-Antonia Martorell-Poveda , José Fernández-Sáez , Mónica Mulet Barberà , Silvia Reverté-Villarroya
- Summary of the manuscript
This manuscript conducts an observational cross-sectional analytical investigation into the psychosocial profiles quality of life and coping methods among anxiety and depression patients in rural Catalonia. The gender-based analysis reveals important information about how different coping strategies affect quality of life and the role that social factors play in mental health outcomes. This investigation adds timely and relevant findings to existing research about gender differences in mental health outcomes.
- General comments
Strengths:
This research fills a significant void in understanding gender differences within mental health disciplines.
The study features a thorough methodology that combines precise sampling methods with validated research instruments.
The presented data validates the conclusions which deliver practical guidance for healthcare professionals.
Areas for improvement:
Further explanation is needed to better justify the study's methodological choices and to provide clearer interpretations of the results.
The paper requires more substantial examination of existing literature on intersectionality and mental health topics.
- Detailed comments
- Introduction
Page 2, Lines 35-39: The introduction successfully outlines global anxiety and depression statistics but needs to incorporate newer data for better relevance. The claims presented would be more credible with additional references from studies published after 2020.
Page 3, Lines 63-69: The research touches on intersectional factors affecting mental health outcomes but needs to expand on this topic. A thorough analysis will improve both the clarity and academic strength of the findings.
- Methods
Page 4, Lines 111-114: The non-probabilistic sampling method needs further justification. Explain the rationale behind choosing this method and identify any possible biases associated with it.
Page 5, Lines 142-148: Elaborate on the measurement methodology used for time-related variables including subjective activities such as personal self-care.
- Statistical analysis
Page 6, Lines 217-234: The statistical approach stands strong yet requires an explanation of why researchers selected the Mann-Whitney U test as opposed to other non-parametric tests for gender-based comparisons.
It is important to add further analysis to properly address potential confounding factors like age and socioeconomic status.
- Results
Page 8, Lines 236-246: The study's findings about socioeconomic disparities between genders stand out but need better integration with the overall objectives addressing gender disparities.
Table 1 (Page 8-9): The salary categories require additional clarification through a short explanation for selecting these particular salary ranges.
- Discussion
Page 13, Lines 367-372: The discussion recognizes gender differences in coping mechanisms yet lacks a thorough exploration of their origins through psychological or sociological theoretical frameworks.
Page 15, Lines 453-459: The under-diagnosis of male depression is mentioned. Support this argument with up-to-date empirical evidence.
- Figures and tables
Figure 1 (Page 10): The graphic representation of statistical significance would benefit from the addition of error bars to enhance clarity.
Table 2 (Page 10-11): The table needs clearer column labels especially in sections indicating statistical significance.
- Ethical considerations
The research document outlines both ethical standards and committee approval details. The authors should provide a more explicit conflict-of-interest statement to guarantee transparency.
- Recommendations for improvement
Clarify sampling justification: The paper should include detailed reasoning for selecting the sampling method used for this study population (Page 4, Lines 111-114).
Expand intersectional analysis: The introduction and discussion require a more thorough examination of how gender interacts with other social determinants of health (Page 3, Lines 63-69).
Elaborate on methodological rationale: Provide a clear explanation for the choice of statistical tests according to the specified page and lines.
Enhance data presentation: revise table and figure legends to improve clarity and precision while making information understandable for readers who lack contextual knowledge (Table 2, Page 10-11).
Strengthen literature engagement: Incorporate updated scholarly research to create a wider contextual understanding of the findings which focuses on the under-diagnosis of males and gender-specific coping mechanisms.
- Overall recommendation
The manuscript presents valuable insights through its wellorganized structure. The paper lacks clear methodological justifications in some areas and its literature discussion does not reach sufficient depth. The study's academic rigor and impact will be strengthened by resolving these identified concerns.
- Final comments
I praise the authors for choosing to explore a topic that holds great social importance and significance. This research paper meaningfully enhances the mental health inequalities discourse through its focus on gender differences in coping mechanisms and quality of life measures. This paper has the potential to become an essential resource for mental health practitioners and researchers after additional refinement.
Yours truly,
Serving peer reviewer at MDPI HEALTHCARE

The manuscript exhibits strong English language quality through its explicit presentation of essential concepts and research outcomes. Some sections need minor adjustments to improve their clarity and flow. The manuscript would benefit from sentence simplification and restructuring to enhance readability along with smoother transitions between sections. The authors should undertake a comprehensive review of the language used in their paper to refine sentence structure and select precise words that help convey the study's scientific contributions with maximum effectiveness.
Author Response
Dear Reviewer 3,
We sincerely appreciate the time and effort you have dedicated to reviewing our manuscript. Your insightful comments and suggestions have been extremely valuable in improving the quality and clarity of our work.
Below, we provide detailed responses to each of your observations, addressing your concerns and implementing the necessary revisions accordingly.
We are grateful for your constructive feedback and hope that the modifications enhance the manuscript to your satisfaction.
Comment 1: Introduction. Page 2, Lines 35-39: The introduction successfully outlines global anxiety and depression statistics but needs to incorporate newer data for better relevance. The claims presented would be more credible with additional references from studies published after 2020.
Response 1: Thank you very much for your contribution. Following your advice, the data has been updated with the latest information published by the WHO on its website (line 44). Thank you.
Comment 2: Page 3, Lines 63-69: The research touches on intersectional factors affecting mental health outcomes but needs to expand on this topic. A thorough analysis will improve both the clarity and academic strength of the findings.
Response 2: Thank you for your valuable feedback. We have tried to highlight the importance of the topic in the introduction. However, we understand that this may not be the main topic, and we did not want to delve deeper into it too much. However, we have added additional text to further emphasize this phenomenon (rows 53 and 122). We hope that these revisions help to better convey the relevance and impact of our research.
Comment 3: Methods. Page 4, Lines 111-114: The non-probabilistic sampling method needs further justification. Explain the rationale behind choosing this method and identify any possible biases associated with it.
Response 3: Thank you for your insightful comment. Regarding the non-probabilistic sampling method, we would like to clarify that all listed individuals were contacted. This list was obtained by identifying all diagnosed individuals who were users of the healthcare center where the study was conducted. For this reason, a simple random sample was not used, as our approach aimed to include the entire eligible population within the study setting. We have included information in section 2.2 to clarify it (lines 157-159).
Comment 4: Page 5, Lines 142-148: Elaborate on the measurement methodology used for time-related variables including subjective activities such as personal self-care.
Response 4: The main questions in this ad hoc questionnaire (marital status, educational level, economic activity) have response options based on previously consulted questionnaires that establish specific options (National Institute of Statistics, 2020; Servicio Andaluz de Salud, 2010). This generated a series of equivalent variables that were easily comparable with similar studies and the national parameters considered. An example of this is the creation of ranges based on the Interprofessional Minimum Wage specified in Royal Decree 231/2020 of February 4 (National Institute of Statistics, 2020), established in Spain at €950 for a 40-hour workweek. Based on this value, ranges were established, according to deciles, to reflect the average wages observed in recent national surveys. Information was also collected on time use outside of work: housework, leisure activities, personal care, care of dependents, and physical activity. To create these variables, we consulted reference sources that describe the conceptual frameworks of contextual determinants of health (Daponte Codina et al, 2019; Martínez Ortega, 2019) as well as institutions such as the World Health Organization (World Health Assembly, 2009). The variables and response options related to these uses of time were configured following what was described in the consulted bibliography (Casado Mejía & García-Carpintero Muñoz, 2018; Eurostat, 2019; Matud, 2015; Velasco Arias, 2009). Based on this literature, the following time ranges were established: <15 minutes per day; between 15 minutes and 1 hour per day; between 1 hour and 2 hours per day; and >2 hours per day. For each of the time-use variables, the tasks or activities covered by each item were specified in the questionnaire itself.
The ad hoc questionnaire also described physical activity levels, which were defined according to the international measurement system extracted from the International Physical Activity Questionnaire (IPAQ), used by national surveys (Ministry of Health, Consumer Affairs, and Social Welfare, 2017). The response options were: low weekly physical activity (little activity, sedentary lifestyle), moderate (walking daily / performing moderate activity on 3 or more days for at least 20-30 minutes/day), or high (vigorous-intensity activities at least 3 days / medium-intensity activities every day).
Casado Mejía, R., & García-Carpintero Muñoz, M. (2018). Género y Salud: Apuntes para comprender las desigualdades y la violencia basadas en el género y sus repercusiones en la salud. Ediciones Díaz de Santos. https://www.editdiazdesantos.com/wwwdat/pdf/9788490521281.pdf
Daponte Codina, A., Cabrera-León, A., Mateo Rodríguez, I., Espinosa de los Monteros, E., Arroyo-Borrell, E., Saez, M., Renart, G., Saurina, C., Serra, L., Bartoll, X., Bravo, M., Domínguez-Berjón, M., López, M., Álvarez-Dardet, C., Marí-Dell’Olmo, M., Bolívar Muñoz, J., Escribà-Agüir, V., Palència, L., Puig, V., … Rueda, M. (2019). Atlas de los determinantes sociales de la salud en España 2019: evolución y variabilidad entre las Comunidades Autónomas (Escuela Andaluza de Salud Pública (ed.)). Publicaciones EASP. https://www.osman.es/wp-content/uploads/2020/03/EASP_Atlas_Determinantes_Salud_Espana.pdf
Eurostat. (2019). The life of women and men in Europe. https://ec.europa.eu/eurostat/cache/infographs/womenmen_2019/
Martínez Ortega, R. M. (2019). Atención y cuidados profesionales a mujeres víctimas de Violencia de Género. FUDEN.
Matud, M. P. (2015). Usos del tiempo, salud y bienestar de mujeres y hombres. In E. Cifre & M. C. Pastor (Eds.), Salud, emociones y género (pp. 91–105). Universitat Jaume I.
Ministry of Health, Consumer Affairs, and Social Welfare. (2017). Encuesta Nacional de Salud de España ENSE 2017: Serie informes monográficos. https://www.mscbs.gob.es/estadEstudios/estadisticas/encuestaNacional/encuesta2017.htm
National Institute of Statistics. (2020). Determinantes de salud. https://www.ine.es/dynt3/inebase/es/index.htm?type=pcaxis&path=/t15/p419/a2017/p06/&file=pcaxis
Servicio Andaluz de Salud. (2010). Escala de calidad de vida WHOQOL-BREF. Programas de Tratamiento Asertivo Comunitario en Andalucía. Documento Marco. http://www.sspa.juntadeandalucia.es/servicioandaluzdesalud
Velasco Arias, S. (2009). Sexos, género y salud: Teoría y métodos para la práctica clínica y programas de salud. Minerva.
Comment 5: Statistical analysis. Page 6, Lines 217-234: The statistical approach stands strong yet requires an explanation of why researchers selected the Mann-Whitney U test as opposed to other non-parametric tests for gender-based comparisons.
Response 5: Thank you for your comment, we kindly try to response your question. Parametric methods work with parameters calculated from the sample. In order to compare groups, in addition to normality, homoscedasticity or equal variances are required, which it results difficulty to justify a priori. In addition, precisely in Health Sciences, the variables tend not to be distributed according to a normal distribution (which does happen in nature) because the work of the health professionals favours the variables "shifting" towards the "good" and deviate from the normal distribution.
On the other hand, this work did not have the objective of comparing the means of the variables as the parametric Student's t test does. Our purpose is to see if the variables are distributed in a different way and for that it is better to use the non-parametric tests because these are based on the order or range of the data.
In the event that the variables are distributed according to a normal distribution, the parametric and non-parametric tests usually reach the same result. In the event that the normality of the variables is not met, non-parametric tests are more accurate since they do not depend on parameters (statistics calculated from the sample) and therefore are not as sensitive to extreme data.
Comment 6: It is important to add further analysis to properly address potential confounding factors like age and socioeconomic status.
Response 6: Dear author, we really appreciate that you have noticed this detail since at the time of writing we made a big mistake in the manuscript. To avoid confounding variables, the OR was adjusted for age (NOT SEX), marital status, SALARY, education and employment status. The error has been corrected and can be seen on line 258.
Comment 7: Results. Page 8, Lines 236-246: The study's findings about socioeconomic disparities between genders stand out but need better integration with the overall objectives addressing gender disparities.
Response 7: The paper describes socioeconomic disparities between genders as a resource for developing a vulnerability profile, and the subsequent achievement of one of the proposed objectives. However, the authors have opted for a less in-depth analysis, allowing for a more comprehensive analysis of all the results obtained.
Comment 8: Table 1 (Page 8-9): The salary categories require additional clarification through a short explanation for selecting these particular salary ranges.
Response 8: Dear reviewer, we consider we have answered this question in response 4.
Comment 9: Discussion. Page 13, Lines 367-372: The discussion recognizes gender differences in coping mechanisms yet lacks a thorough exploration of their origins through psychological or sociological theoretical frameworks.
Response 10: Thank you for your insightful comment. We acknowledge the importance of exploring the psychological and sociological theoretical frameworks behind gender differences in coping mechanisms. However, we opted not to expand further in this area due to the already extensive length of the discussion section. Additionally, delving deeper into these theoretical frameworks would not significantly enhance the value of our study as it is currently structured. We appreciate your thoughtful feedback and remain open to providing further clarifications if needed.
Comment 10: Page 15, Lines 453-459: The under-diagnosis of male depression is mentioned. Support this argument with up-to-date empirical evidence.
Response 10: Thank you for pointing this out. We have added a reference to a recent article analyzing this phenomenon. This mention is thus supported by several references to studies published between 2016 and 2022 (line 526).
Comment 11: Figures and tables
Figure 1 (Page 10): The graphic representation of statistical significance would benefit from the addition of error bars to enhance clarity.
Response 11: Thank you for your valuable suggestion. During the development of the manuscript, we considered incorporating graphical representations; however, we ultimately decided to maintain uniformity and ensure consistency in the presentation of the data throughout the article. We appreciate your feedback and remain open to any further clarifications if needed.
Comment 12: Table 2 (Page 10-11): The table needs clearer column labels especially in sections indicating statistical significance.
Response 12: In response to your comment, the authors have deemed it appropriate to modify the description of the stratification of results in lines 331-333, and modify the Table 2 legend (line 373). We hope this will make the table easier to interpret without the need for modification, thus maintaining the uniformity of the manuscript.
Comment 13: Ethical considerations. The research document outlines both ethical standards and committee approval details. The authors should provide a more explicit conflict-of-interest statement to guarantee transparency.
Response 13: Thank you for your observation. Regarding the conflict-of-interest statement, we have followed the journal’s author guidelines to ensure compliance with its requirements. Additionally, in the ethical considerations section, we have provided detailed information, which is also explicitly stated in the "Institutional Review Board Statement" section (line 664). Furthermore, the favourable opinion from our ethics committee encompasses all these aspects and has been officially recorded under protocol code 20/157-P, which is unique to this study.
To ensure that we have not forgotten any of your comments, we have considered it appropriate to also respond to your final "Recommendations for improvement" in order to provide you with a complete response to your report:
Comment 14: Clarify sampling justification: The paper should include detailed reasoning for selecting the sampling method used for this study population (Page 4, Lines 111-114).
Response 14: Thank you for your advice. We have answered this question in Response 4 and, as indicated, we have included information in point 2.2 to clarify this (lines 157-159).
Comment 15: Expand intersectional analysis: The introduction and discussion require a more thorough examination of how gender interacts with other social determinants of health (Page 3, Lines 63-69).
Response 15: Thank you for your advice. We have answered this question in Response 2.
Comment 16: Elaborate on methodological rationale: Provide a clear explanation for the choice of statistical tests according to the specified page and lines.
Response 16: Thank you for your advice. We have answered this question in Response 5.
Comment 17: Enhance data presentation: revise table and figure legends to improve clarity and precision while making information understandable for readers who lack contextual knowledge (Table 2, Page 10-11).
Response 17: Thank you for your advice. We have answered this question in Responses 11 and 12.
Comment 18: Strengthen literature engagement: Incorporate updated scholarly research to create a wider contextual understanding of the findings which focuses on the under-diagnosis of males and gender-specific coping mechanisms.
Response 18: Thank you for your advice. We have answered this question in Response 10.
The authors greatly appreciate your feedback and believe your comments have greatly improved the original manuscript. Thank you very much for your work.
Sincerely,
The authors.